# UNDER WHAT CIRCUMSTANCES DO LOCAL CODES EMERGE IN FEED-FORWARD NEURAL NETWORKS.

## ABSTRACT

Localist coding schemes are more easily interpretable than the distributed schemes but generally believed to be biologically implausible. Recent results have found highly selective units and object detectors in NNs that are indicative of local codes (LCs). Here we undertake a constructionist study on feed-forward NNs and find LCs emerging in response to invariant features, and this finding is robust until the invariant feature is perturbed by 40%. Decreasing the number of input data, increasing the relative weight of the invariant features and large values of dropout all increase the number of LCs. Longer training times increase the number of LCs and the turning point of the LC-epoch curve correlates well with the point at which NNs reach 90-100% on both test and training accuracy. Pseudo-deep networks (2 hidden layers) which have many LCs lose them when common aspects of deep-NN research are applied (large training data, ReLU activations, early stopping on training accuracy and softmax), suggesting that LCs may not be found in deep-NNs. Switching to more biologically feasible constraints (sigmoidal activation functions, longer training times, dropout, activation noise) increases the number of LCs. If LCs are not found in the feed-forward classification layers of modern deep-CNNs these data suggest this could either be caused by a lack of (moderately) invariant features being passed to the fully connected layers or due to the choice of training conditions and architecture. Should the interpretability and resilience to noise of LCs be required, this work suggests how to tune a NN so they emerge.

## 1 INTRODUCTION

With neural networks (NNs) being widely deployed in various tasks it is essential to understand how they work and what data is used to make their decisions. NNs used to be viewed as 'black boxes', but recent results (Nguyen et al., 2016) have started to open that box. NNs came from the field of psychology as simple bio-inspired models, it has been debated whether information is represented in the brain in a distributed manner (from parallel distributed processing, PDP) or via a localist coding scheme. Although the distributed approach was the most popular, there are some results in neuroscience (Quiroga et al., 2005) and psychology (McClelland & Rumelhart, 1981) that are commensurate with a localist coding scheme, including a report of LCs in RNNs (Bowers et al., 2014). Recently, there has been an explosion of interest in NNs, especially deep-NNs, as these algorithms are now commercially relevant, and this increase in their accuracy has been credited to sources of vastly more labelled data and novel training techniques like dropout (Srivastava et al., 2014). Many newer researchers in NNs were perhaps unaware of the distributed-localist coding debate within psychology, and thus looked for localist-like codes in their NNs, and found indicative (of LC coding scheme) evidence of detectors for objects (Zhou et al., 2018; 2015), concepts (Karpathy et al., 2016; Lakretz et al., 2019), features (Nguyen et al., 2019; Erhan et al., 2009), textures (Olah et al., 2017), single directions (Morcos et al., 2018) etc., see (Bowers, 2017) for a review.

With faster and larger computers, it is possible, even with the increase in input data size, for deep-NNs to 'memorise' the data-set (an extreme form of overfitting): a process where the NN has simply learned a mapping between input and output vectors, as opposed to learning a rule which will allow it to generalise to unseen data that follows the underlying rule. Generalisation performance is often improved if NN training is stopped early, often when a validation set loss (val_loss) stops improving,

as the NN is prevented from further minimising its loss function by memorising the input (training) data. Single directions (Morcos et al., 2018) have been implicated in memorization of the data-set.

Localist codes (coding for a class $A$) are defined as units which are activated at a high (low) level for all members(that the NN gets correct) and low (high) level for all members of the other classes (class $\neg A$), i.e. the set of activations for class $A$ is disjoint from the activations for class $\neg A$ (see figure 8 in the appendix), and these codes are very strict measure of selectivity. As such, LCs are very easy to interpret, and the presence of them in NNs would make it easy to understand how the NN is working.

This paper takes no position on whether or not localist codes exist in the brain or in deep-NNs. instead we take the constructionist science approach of asking when would we expect LCs to appear, and what aspects of the system, data-set and training conditions favour or disfavour their emergence. As NNs are considered (simplified) models for the brain, we can also take into account biological plausibility. We hypothesized that LCs should emerge when there was an invariant in the data-set. As deep-NNs take a long time to train, it is hard to get representative statistics, so we look at very simple networks (shallow: 3-layer and pseudo-deep: 4-layer) where it is possible to do hundreds of repeats and thus get resilient trends.

The main insight of this work is that LCs do emerge when there is an invariant in the data. To set up a system with such a 'short-cut' we use a simple binary vectors as inputs, which are built from prototypes, such that there are 1/10 input bits that are always 1 for each class and these are the invariant bits, the 0s of each prototype are then filled in with a random mix of 1 and 0 of a known weight, see figure 1, thus, a given bit is always on for a given class, and maybe on or of for other classes. Note also, that in this set up, if the proportional weight of the prototype exceeds that of the random vector, then vectors belonging to the same class are 'closer'[1] to each other than those of separate classes, i.e. there is a larger between-class variance than within-class variance. The prototypes are also perturbed to increase the variance of the 'invariant' bits. If one views a deep conv-NN as a feature extraction machine (lower and convolutional layers) with a feature classification NN on top (the higher fully connected layers), then it is reasonable to suppose that a given class is likely to share features at the top convolutional layer, which would result in the activation vectors at that layer having a higher between group variance than within group, or possibly even invariant features for a class (perhaps object detectors), and so these experiments could give insight into the representation of data in the 'fc' layers of deep-NNs.

## 1.1 FINDINGS

1. LCs related to lower within-class variance than between-class variance in input
2. LCs related to a NN internalising a rule
3. No. of LCs related to difficulty of the problem and the computing power of the NN, with different behaviour for under- and over-resourced NNs.
4. Large values of dropout increases LCs
5. LCs correlate with generalisation performance
6. Large data-sets, softmax and aggressive early stopping reduce the number of LCs
7. Monitoring the number of LCs can be useful for figuring out when to stop training

## 2 METHODOLOGY

**Data design**  Data input to a neural network can be understood as a code, $\{C_x\}$, with each trained input data vector designated as a codeword, $C_x$. The size of the code is related to the number of codewords (i.e. the size of the training set), $n_x$. $L_x$ is the length of the codeword, generally 500bits in this paper. We used a binary alphabet, and the number of 1s in a codeword is the weight[2], $w_x$ of that codeword.

---

[1]There are two metrics that are relevant to measuring the distance between these vectors, the Hamming distance which is the number of bits that have to be switched to turn one vector into another and the cosine similarity, which is the angle between the vectors, we use Hamming distance here.

[2]This weight definition is not the same as connection weights in the neural network.

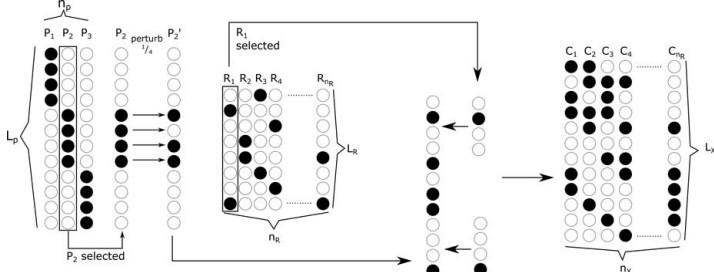

Figure 1: Schematic for building a random code with known properties. Black circles represent ones, white circles represent zeros. Class prototypes ($P_1$, $P_2$, and $P_3$) were made with length $L_P$; the number of prototypes are $n_p$, which is three in this example, their weight is four, with a sparseness number, $S_p$ of $\frac{1}{3}$. Random vectors, $R_x$, were made, as shown, these have length $L_R$ and there are $n_R$ of them; they have weight, $w_R$ of two and sparseness number, $S_R$, of $\frac{1}{4}$. To assemble a new codeword, a prototype is chosen, in this example, $P_2$, and 'perturbation' errors are applied, in this example, the perturbation rate is $\frac{1}{4}$, so a single one is turned to a zero in the modified prototype ($P_2'$). A random vector, in this example $R_1$, is then generated, split into blocks and added to the parts of the prototype ($P_1$) that were zero (the random vectors cannot overwrite a random decay in $P$). The process is repeated to create an input 'code' with $n_x$ codewords of length $L_x$, where $n_x = n_R$ and $L_x = L_P$. If the decay values is sufficiently low, members of each class, as they are based on the same prototype are more similar to each other than codewords in other classes.

To create a set of $n_P$ classes with a known structural similarity, the procedure in figure 1 was followed. We start with a set of $n_P$ prototypes, $\{P_x, 1 \le x \le n_P\}$, with blocks of 1s of length $L_P/n_P$, called prototype blocks, which code for a class. For example, if $L_x$ were 12 and $n_P$ were 3: $P_1 = [111100000000]$ and $P_2 = [000011110000]$, $P_3 = [000000001111]$, and this would gives prototypes that are a Hamming distance of 8 apart, and thus we know that our prototypes span input-data space. To create members of each class, the prototype is used as a mask, with the 0 blocks replaced by blocks from a random vector, $R_x$. The weight of the random vectors, $w_R$ can be tuned to ensure that a set of vectors randomly clustered around the prototype vector are generated, such that members of the same category are closer to each other than those of the other categories (N.B. the prototypes are not members of the category). A more realistic data-set is created by allowing the prototypes to be perturbed so that a percentage of the prototype block is randomly switched to 0s each time a new codeword is created, in accordance with the perturbation rate (see $P_2'$ in figure 1). This method creates a code with a known number of invariant bits for codewords in the same category. For example, in figure 1, codewords $C_1$ and $C_2$ were both derived from $P_1$, and have a Hamming distance of 6, where as $C_1$ and $C_4$ are in different classes and have a Hamming distance of 8. Note, the difference between these numbers is larger in our experiments as $L_x = 500$ and there are 10 categories. We define 'sparseness' of a vector, $S_x$, as the fraction of bits that are '1's. LCs were highly unlikely to appear in the input code, and none were observed in random checks.

**Neural network design**  We have three-layer feed-forward network with $L_x$ input neurons, $n_{HLN}$ hidden layer neurons (HLN) and $L_o$ output neurons, using a sigmoidal activation function and no softmax on the output. For experiment 1 $L_x$ is 500bits, mapped to 10 output classes, so the weight of prototype vectors $w_p$, is 50, and the $w_R$ is 150 (so $S_r$=1/3), the output vector is a 50bit-long distributed vector ($w_o$=25). NNs were trained for 45,000 epochs, and each plotted point is a number of repeats between 10 and 15. Experiment 2-7 varied thus: 2: $n_x = \{250, 500\}$; 3: $S_R = \{^1/_9, ^2/_9, ^1/_3\}$; 4: $L_x = \{300, 700, 1000\}$; 5: activation function=\{ReLU, sigmoid\}; 6: output vector is distributed or 1-hot; 7: takes $n_{HLN} = 500, 1000$ and decays the $w_P$ from 50 to 25 ($P$ from 1 to 0.5). Experiment 8 is $\sim$250 repeats of experiment 1, with values of dropout in $\{0, 0.2, 0.5, 0.7, 0.9\}$. Experiment 9 is a repeat of 1, with activation noise added for networks with $n_{HLN} = \{100, 500, 1000\}$. Experiment 10 measures the number of LCs over training time for $n_{HLN} = \{250, 500, 1000, 2000\}$. To do generalisation tests, a new test set is built with the same parameters as the training set, with $n_{train} = 10000$ and applied to pre-run results (from experiments 8 and 10). Experiment 11: for the 4-layer neural networks, $n_{HLN}$ of the first hidden layer is varied, the second is set to 250, the output vectors are 1-HOT and different training parameters and values are given in table 11. Experiment

12: For the MLP experiments, we use MNIST data-set, with added 20 pixel invariants that code for the class which are either non-varying ('invariant') or drawn from a Gaussian distribution ('Gaussian'), see table 8 in the appendix. The invariant was either not applied ('standard') or applied to 50% of 100% of all images, or applied to the whole of 2, 5 or 8 categories, and data is from 10 repeats. Experiment 13: To see if LCs were associated with memorising the dataset, we trained NNs where the codewords were shuffled before being assigned to targets. NNs were run in Keras with a TensorFlow backend.

| Experiment | Activation function | Softmax | Dropout 20% | Size of training set | Stopping condition |
|---|---|---|---|---|---|
| A | sigmoid | no | yes | $n_x = 100$ | 60,000 steps |
| B | sigmoid | no | no | $n_x = 100$ | validataion set loss |
| C | sigmoid | yes | no | $n_x = 100$ | 60,000 steps |
| D | sigmoid | yes | yes | $n_x = 100$ | validation set loss |
| E | sigmoid | no | no | $n_x = 100$ | training accuracy |
| F | sigmoid | yes | no | $n_x = 100$ | training accuracy |
| G | ReLU | yes | no | $n_x = 100$ | validation set loss |
| H | ReLU | yes | no | $n_x = 100$ | 60,000 steps |
| I | ReLU | yes | no | $n_x = 100$ | training accuracy |
| J | ReLU | no | yes | $n_x = 100$ | 60,000 steps |
| K | ReLU | no | no | $n_x = 100$ | validation set loss |
| L | ReLU | no | no | $n_x = 100$ | training accuracy |
| M | ReLU | no | no | $n_x = 1000$ | 60,000 steps |
| N | ReLU | no | no | $n_x = 100$ | 60,000 steps |
| O-S | ReLU | * | * | $n_x = 1000$ | * |

Table 1: Training conditions for experiment 11. *O-S included tests for softmax, dropout and the three training conditions.

**Accuracy and generalisation.** The accuracy reported by Keras counts an output vector as correct if each bit is within 50% of the correct value, e.g.. the output vector [0.4, 0.6] would map to the target vector [0, 1] (the outputs are binary), and in standard classification neural networks with a 1-hot target vectors these outputs would be very close to the target after the softmax operation. Thus, we label this accuracy 'classification accuracy'. If one wants to use a NN as a pattern matching machine, then one could put an arbitrary limit on how big an error between the output values and the targets, we chose 10%, and the target [0, 1] would need an output vector of [$\leq 0.1$, $\geq 0.9$] to be considered correct. We call this more stringent condition the 'pattern matching accuracy.' As the codes are built to a rule, it is possible to generate an arbitrarily large code, thus our test sets are at least 10 times larger than the training sets.

## 3  RESULTS

As the chance that all the members of $A$ would emerge disjoint from the members of $\neg A$ is $\binom{50}{50}/\binom{500}{50}$ is tiny ($4.32 \times 10^{-71}$), finding a single interpretable local code, such as those shown in figure 8 in the appendix, refutes the idea that neural networks do not have interpretable or locally encoded units.

**Local codes are seen in response to an invariant** As shown in fig. 2a, the number of local codes is tuned by the size the hidden layer, with a peak in the number of local codes seen at $n_{HLN}$=1000 for the standard data set, and a peak in the percentage of HLNs which are local codes seen at $n_{HLN} = 500$ (data not shown): dashed and dot-dashed grey lines are drawn at these points in all relevant figures. The shape of the graph (in fig.2) appears to be an interaction between the difficulty of the problem and the amount of computing power (i.e. the $n_{HLN}$) available, with those networks to the left of the peak ('under-powered' with respect to LC emergence NNs) not having enough spare units to give over to a localist code, and those to the right ('over-powered'). Note that, in the results that follow, the data for $n_{HLN}$ is often qualitatively different for over- and under-powered NNs; for example, the number of LCs decreases over time for longer times or the underpowered NNs in experiment 10 and figure 3a and increases for over-power NNs.

Increasing the difficulty of the problem (by increasing $n_x$, see fig.2a, increasing $L_x$ see fig. 9 in appendix, or decreasing the relative information in the prototype by decreasing sparsity of $S_R$, see fig. 2b decreases the number of local codes. The proportionately larger the invariant is in the input code, the more likely LCs are to emerge in the NN, presumably as a 'rule' that attends only to the invariant is correspondingly more efficient. In the MLP experiments, the presence of invariant features in pixel space did cause a small number of LCs to emerge[3] (see table 9 in the appendix), and these systems trained much faster (these were trained to a given accuracy), see tables 10 in the appendix).

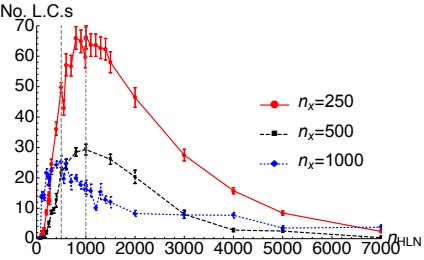

(a) Experiment 1 and 2: The number of local codes (LCs) against the number of HLNs, $n_{HLN}$, for different numbers of training examples ($n_x$).

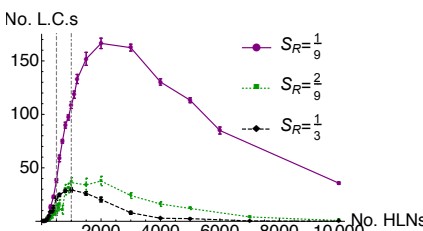

(b) Experiment 3: The number of local codes against $n_{HLN}$ for varying the sparseness of the random blocks of the vector, $S_R$.

Figure 2: Input data that is few in number and sparse exhibits more local codes. As the prototype vector in all these cases has a sparseness of $^1/_{10}$, the weight of the random $w_R$ and prototype, $w_P$, parts of the codeword and the codewords are 500bits long, note that purple: $S_x = 0.2$, $w_R$=50, $w_P = 50$; green: $S_x = 0.3$, $w_R$=100, $w_P = 50$; black dashed: $S_x = 0.4$, $w_R$=150, $w_P = 50$ Gray dashed and dot-dashed lines are drawn at $n_{HLN}$=500 and 1000 respectfully.

**LCs increase with Training time and may relate to generalisation**  Figure 3a shows that the number of LCs in a NN increases with training time all across the range of $n_{HLN}$. The corner of the LC-epoch curve (Fig. 3b) was roughly correlated with the system achieved 100% classification accuracy ('easy' criteria) on the training data and the asymptote on the test data (around 95%), see figure 3c. Interestingly, this point is also when the pattern-matching accuracy ('hard' criteria) on both the train and test data-sets rises above 0 (fig.3d, suggesting that the additional LCs emerging are caused by the separation of 1s and 0s in the NN output. Thus, monitoring the rate of LC increase can give a measure of when to stop training.

To see if LCs were related to memorisation, we investigated the experiment with the greatest range in LCs-the 90% dropout shown in figure 7b-and the training and test accuracy is plotted in fig. 4a. There were many solutions that had an accuracy $> 99\%$, but where the solutions were lower accuracy, the accuracy on both train and test data was related to the number of LCs. Note that, as the test set was many times larger than the training set ($n_x$= 10,000 for test, $n_x = 500$ for training) this shows that the NNs have clearly internalised the rule, the rough positive correlation between the number of LCs and the accuracy (in the accuracy range of 20-100%) demonstrates that LCs are part of the internalisation of this rule. Similarly, plotting the test accuracy against the number of LCs for experiment 10 (fig.4b) shows a positive correlation between the no. of LCs and classification accuracy (similar results are seen with the pattern-matching accuracy, see figure 12 in the appendix). Shuffling the input code to force the NN to memorise the data resulted in a reduction in the number of LCs, but only by a maximum 47% (see fig. 13 in the appendix).

**Why might LCs not be seen in modern deep NNs?**  Figure 5 shows the effect of changing the system setup and output vectors. There is little difference between ReLU and sigmoid, except that the peak of the curve is shifted a little to the left in the ReLU data, likely as ReLU is a more 'powerful'[4] activation function than sigmoid. An alternative explanation is that as ReLUs train faster and the number of local codes increases with training time (see fig.3a), the ReLU networks are

---

[3]No LCs were seen in the standard MNIST runs.

[4]i.e. its range is not limited at the positive end so can use a larger number space than sigmoid

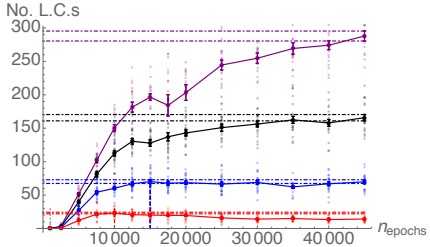

(a) Local codes increase with training time ($n_{epochs}$

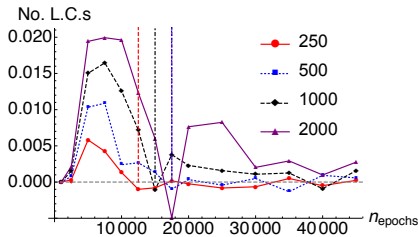

(b) Taking differences to identify the turning points of the curves.

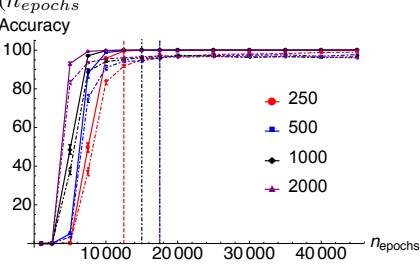

(c) Classification accuracy on train (solid) and test (dashed).

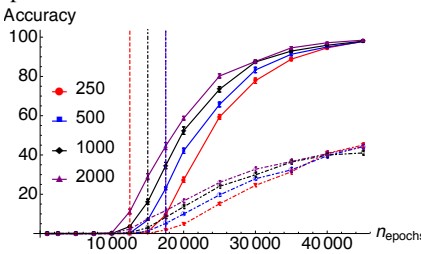

(d) Pattern matching accuracy on train (solid) and test (dashed).

Figure 3: Experiment 10. The number of local codes increases with training time. The point at which the rate of LC code addition decreases correlated with both when the NN achieves ≈99% classification accuracy and when it achieves classification accuracy of more than 0. Key: red: $n_{HLN}$ = 250; blue: $n_{HLN}$ = 500; black: $n_{HLN}$ = 1000; purple: $n_{HLN}$ = 2000.

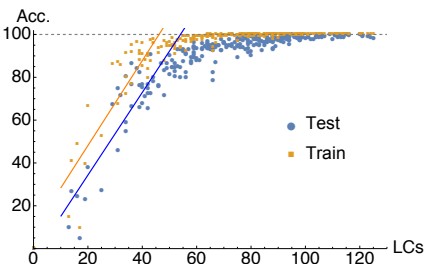

(a) Relation between classification accuracy on test and train for the 90% dropout experiment (8).

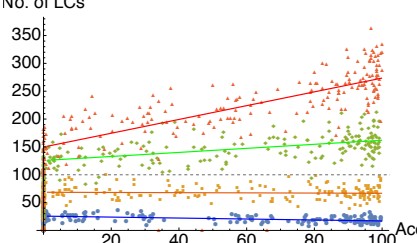

(b) Relationship between the number of LCs and classification accuracy on test from experiment 10.

Figure 4: The number of local codes increases with improvement on generalisation accuracy. Fig.4a is over 250 repeats trained for the same amount of time, Fig.4b are repeated measurements on the same NNs as they are trained.

effectively trained for longer compared to sigmoidal networks. We chose to use a distributed output code in the 3-layer networks as we thought the local codes at the output layer might encourage the emergence of LCs in the hidden layer: the opposite happened, likely because a simple 3-layer network set up with prototypes like this, the problem of mapping input to output is linearly separable and much easier, hence LCs are not needed. Note that, in deeper 4-layer HLNs with 1-hot output vectors there are many local codes (see 'B' in figure 6b).

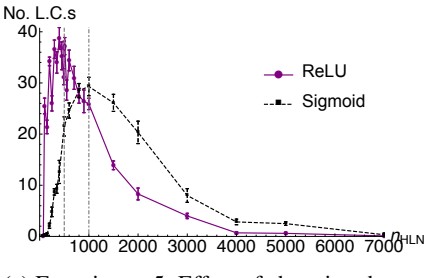

(a) Experiment 5. Effect of changing the activation function.

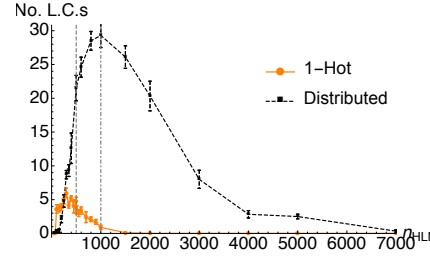

(b) Experiment 6. Effect of changing the output form.

Figure 5: Network architecture parameters can drastically effect the emergence of local codes. 5a: Switching from HLNs with a sigmoidal activation function to a rectified linear (ReLU) units; 5b: Switching from a distributed to a 1-hot output encoding.

To test deep-NNs, we compared 4-layer NNs with our standard set-ups (A, B) that exhibited LCs with similar architecture and standard CNN training techniques added (e.g. L, O-S). NNs with all the settings commonly found in deep-NN literature do not produce LCs. Figure 6a shows that there is little difference between stopping on val_loss or after 60K epochs (as does the overlap between bars A and B in fig. 6b. L, M, O-S had ReLU and large data-sets, which completely inhibited local codes. H to S used ReLU rather than sigmoid, so in these systems it seems that ReLU-based NNs do not learn local codes, although they did in our earlier experiments (fig. 5). That C and D are so similar shows no real effect of dropout in these NNs. That C<A, D<B, F<E (and fig. 6a) shows that softmax reduces the number of local codes slightly. We propose this is because softmax does some extra 'processing' on the final layer, shifting 50% accurate output vectors up to 90%, and this extra processing makes the problem easier for the hidden layers (so the 'rule' is less useful). Alternatively, due to this extra 'processing', using softmax reduces the time required to train the network, which is expected to reduce the number of LCs, see figure 3a, (N.B. with early stopping conditions training time now varied). Thus, the reason that LCs are not found in deep-NNs could be the use of large training data sets, early stopping, ReLU and softmax in those systems, or alternatively it could be because there are no feature invariants associated with output classes at the top convolutional layer (i.e. the layer before the fully connected classification layers).

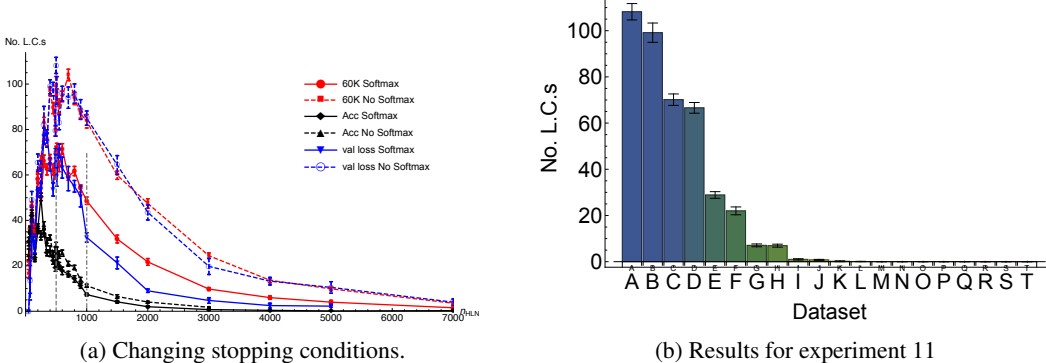

(a) Changing stopping conditions.

(b) Results for experiment 11

Figure 6: 'Pseudo-deep' NNs that exhibit LCs lose them if modern deep-learning training techniques are applied.

### 3.1 BIOLOGICAL CONSTRAINTS ENCOURAGE THE EMERGENCE OF LCs

As the positive activation range of the ReLU is unlimited, it is less biologically plausible than the sigmoid activation function (although these NNs are still very far from being a model of biology). In this section we investigate noise on sigmoidal, non-softmax NNs. As real biological systems are (possibly) noisy[5], we looked a several different ways of introducing noise into the system: decaying the invariant, dropping out units and adding in activation noise.

Perturbing the prototype block is shown in figure 7a, and LCs are still seen if the invariant is not always present, but simply more likely to be present for a given class. However, the number of LCs disappears when the decay is 0.4, i.e. weight of the (50-bit long) prototype section is 20bits, and this is roughly equal to the weight of a 50bit long section of the random vector (16.67bits), as at this point the signal is indistinguishable from the noise. The curves are not well fit by an exponential, the fits shown here are: (for $n_{HLN} = 500$) 20.62 - 30.65 $\sqrt{P}$, with $R^2 = 0.984$; (for $n_{HLN} = 1000$) $19.72 - 32.29\sqrt{x}$ with $R^2 = 0.977$, and these data were fit up to $P=0.4$. Comparing to the MLP case, there were no LCs if the invariant feature was perturbed, see table 9.

Contrary to our expectations, adding dropout and increasing it to very large values increases both the average and maximum number of LCs in solutions found by the neural network, although lower values of dropout seemed to overlap, with two competing possible underlying distributions seen perhaps between 0 and 40% (see fig. 10 in appendix). Summary statistics and Kolmogorov-Smirnov hypothesis tests are reported in tables **??** in the appendix, and above 50% dropout there is a significant difference in the underlying distributions with an increase in number of LCs. Repeated copies of local codes may be more resilient to noise than a distributed code: a fully distributed code will always be incomplete but a local code only has a chance of being not present under dropout.

Adding in activation noise changes the number of LCs that are found (see fig. 11 in the appendix), increasing them in overpowered NNs and decreasing them in underpowered networks.

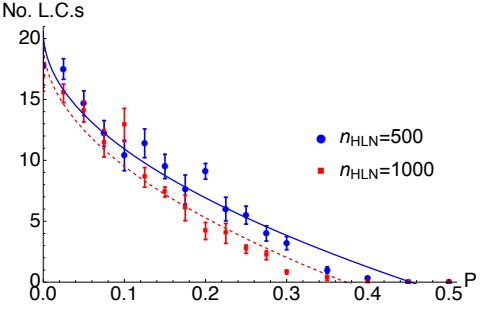
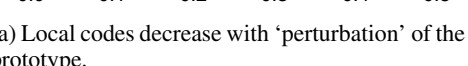

(a) Local codes decrease with 'perturbation' of the prototype.

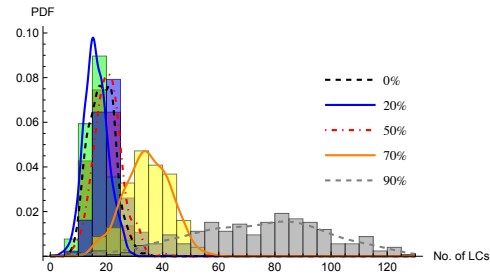

(b) Probability Density Function (PDF) and bar charts for NNs run with different values of dropout (in legend).

Figure 7: How noise affects local codes. Fig.7a: perturbing the prototype part of the codeword decreases the drive to learn local codes. Perturbation, $P$, is measured as the number of bits in prototype block code that are randomly flipped. Fig.7b: increasing dropout increases the number of local codes. As the dropout percentage increases, generally, the mean and range of local codes found increases, suggesting that localized encoding by the network offers some advantage against noise.

## 4 DISCUSSION

The number of local codes is not large, usually between 0-10% of the hidden layer, but this is not insignificant and the fact that they emerge in response to invariance in the data-set and some training conditions suggest that they have a function (presumably detecting the presence of an invariant which can be used as a 'short-cut'). These results are resilient to a small perturbation of the invariant (in the NNs), the invariant feature does not need to be always present, merely more likely

---

[5]It is commonly suggested that living neural networks are noisy and thus that neurons and synapses must be resilient to noise, however there is some debate as to whether this is true and some neurons are highly un-noisy.

to be present for a specific class. LCs emerge in correlation with generalisation performance, so monitoring the rate of change of LC numbers can give feedback on when to stop training to avoid overfitting. As LCs were found in response to an added invariants in 2D graphical data in MLPs, we expect that the use of LCs to identify and code for found 'rules' based on invariant features will be a general phenomenon. Morcos et al. (2018) found that 'single direction' codes were associated with the harder problem of memorising the dataset (as compared to learning the simpler underlying rule), here we found that LCs emerged in response to learning a simpler rule, and were inhibited by switching to a memorisation task. Further work is needed to tease out which aspects of the difference in architecture and problem choice between Morcos et al. (2018)'s study and ours is the cause.

It's an open question whether there are localist codes in deep-CNNs, PDP dogma suggests not, but there have been some highly selective units found in other types of NNs. Our results suggest that there might not be LCs in the fully connected layers if typical current deep-CNN training methods are used. Furthermore, there is the question of whether there are invariant features coming from the conv/pooling layers into the fully connected feed-forward layers (of a typical current deep-CNN) which is not currently known and an area of active research for us.

As LCs were seen when the NN has slightly more biologically plausible training conditions like a noisy system, it might be possible that LCs might emerge in the human brain, although even the more biologically plausible NNs are perhaps too far removed from actual biology.

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

# A  APPENDIX

## A.1  LOCAL CODE EXAMPLES

Jitterplots are common analysis techniques in neuroscience, but can also be applied to neural net-worksBerkeley et al. (1995) by plotting the activation of a unit on the x axis and randomly jittering the y-axis value. Examples of local codes found in our networks are given in figure 8, these were sigmoidal activation units. As it takes more energy to spike than not spike, in biological neurons the localist codes are generally 'on' codes where the unit is on more strongly in the presence of the class it codes for than not, see figure 8left. In our system there is no energy penalty for using high activations (in fact, energetic considerations are not modelled at all, and this is one of the reasons NNs are not biologically plausible) so we also see 'off' codes: see figure 8right.

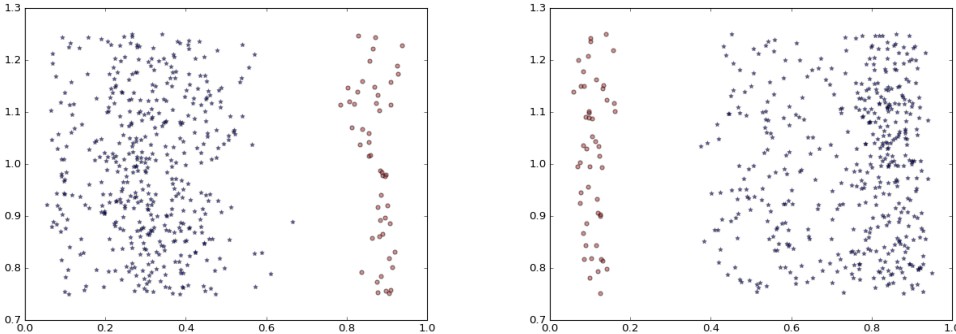

Figure 8: Examples of interpretable local codes found in a distributed network. Left: a selectively on unit with a selectivity of $\sim +0.12$; Right: selectively off unit with a selectivity of $\sim -0.2$. Red circles belong to a single category, blue stars are all the members of all other categories, the x-axis is the activation of a hidden layer neuron (HLN) and points are jittered randomly around 1 on the y-axis for ease of viewing. There is a clear separation between activations for the class depicted in red (A) and all other activations (not-A), thus examination of the activations of these units would reveal the presence or absence of the red class.

## A.2  EXPERIMENT 4: CHANGING INPUT VECTOR LENGTH ($L_x$)

As shown in figure 9, there is little difference between different lengths of vectors for smaller lengths, although the number of LCs is decreased at $L_x = 1000$.

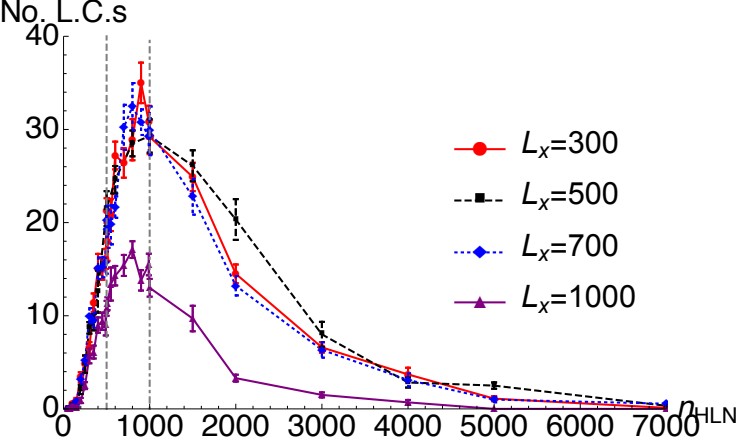

Figure 9: Experiment 4. The effect of changing the input vector length.

## A.3 Further results from experiment 8: dropout

Dropout (see Srivastava et al. (2014)) is a common training technique where a percentage of a layer's neurons are 'dropped out' of the network (their connections are set to zero) during training to prevent over-fitting, however, dropout can also be viewed as a type of training noise. Results are shown in figure **??** and table 2. There are always solutions with close to zero local codes, but the expected (mean) and maximum number roughly increases with an increasing percentage of dropped out neurons.

Generally, dropout percentages in the range of 20-50% are used in training, and these probability distribution functions (PDFs), like 0% peak, are also joint peaks, with the 20% data having more of the lower solution and the 50% having more of the higher one. Dropping out more than 50% of the network is not generally used as it slows down training. However, with these higher values, the range of solutions is much higher (as evidence by a higher variance and range of the number of local codes), which is expected as dropout forces the network to adopt a range of solution sub-networks, the increase in local codes suggest that localised encoding offers some protection against noise.

At first glance, this might seem unlikely, as distributed patterns are claimed to be more resilient against failure. However, say we had a 20% dropout rate, a fully distributed encoding, would be affected by dropout 100% of the time, losing 20% of its information, whereas a localised encoding would be unaffected 80% of the time (although 20% of the time it would lose all data), and further resilience can be provided if duplicate local codes were used for the same class. Note that, as only 10 classes were used, the large number of local codes, especially for the high dropout values, suggests there are multiple LCs for each category. These results suggest that, for a noisy network, solutions involving some duplicate localised codes are useful methods for dealing with uncertainty.

Table 2: Various quantities associated with the distribution of local codes (LCs) in with dropout (given as a percentage) applied during training

| No. of LCs | 0% | 20 % | 50% | 70% | 90% |
|---|---|---|---|---|---|
| Minimum | 8 | 4 | 7 | 15 | 0 |
| Maximum | 34 | 29 | 41 | 57 | 125 |
| Mean | 18.44 | 16.13 | 20.65 | 34.54 | 73.80 |
| Standard deviation | 4.55 | 4.19 | 4.96 | 8.05 | 24.53 |

To test if the different datasets could have come from the same distribution we did Kolmogorov-Smirnov hypothesis tests with the hypothesis being that the data were from the same distribution. As can be in seen tables **??** and figure **??** low values of dropout are likely the same distribution, but the distributions are different for the high values of dropout. The expanded set of experiments is shown in figure 10, and there seems to be two underlying distributions between 0 and 50%, with the 10%, 20% 30% dropout NNS having more of the distribution with less LCs, and 0%, 40% and 50% having more of the distrubtion with more LCs.

| Dropout | Dropout | Result | p-value |
|---|---|---|---|
| 0% | 0% | Do not reject | 1. |
| 0% | 10% | Reject | 0.0000489944 |
| 0% | 20% | Reject | $4.749456190224837 \times 10^{-6}$ |
| 0% | 30% | Reject | 0.0000247196 |
| 0% | 40% | Do not reject | 0.137226 |
| 0% | 50% | Reject | 0.000251174 |
| 0% | 60% | Reject | $2.4240973576630998 \times 10^{-31}$ |
| 0% | 70% | Reject | $1.3379086295029856 \times 10^{-80}$ |
| 0% | 90% | Reject | $5.31281662567248 \times 10^{-122}$ |

Table 3: Kolmogorov-Smirnov hypothesis tests for the dropout experiments compared to a run with no dropout.

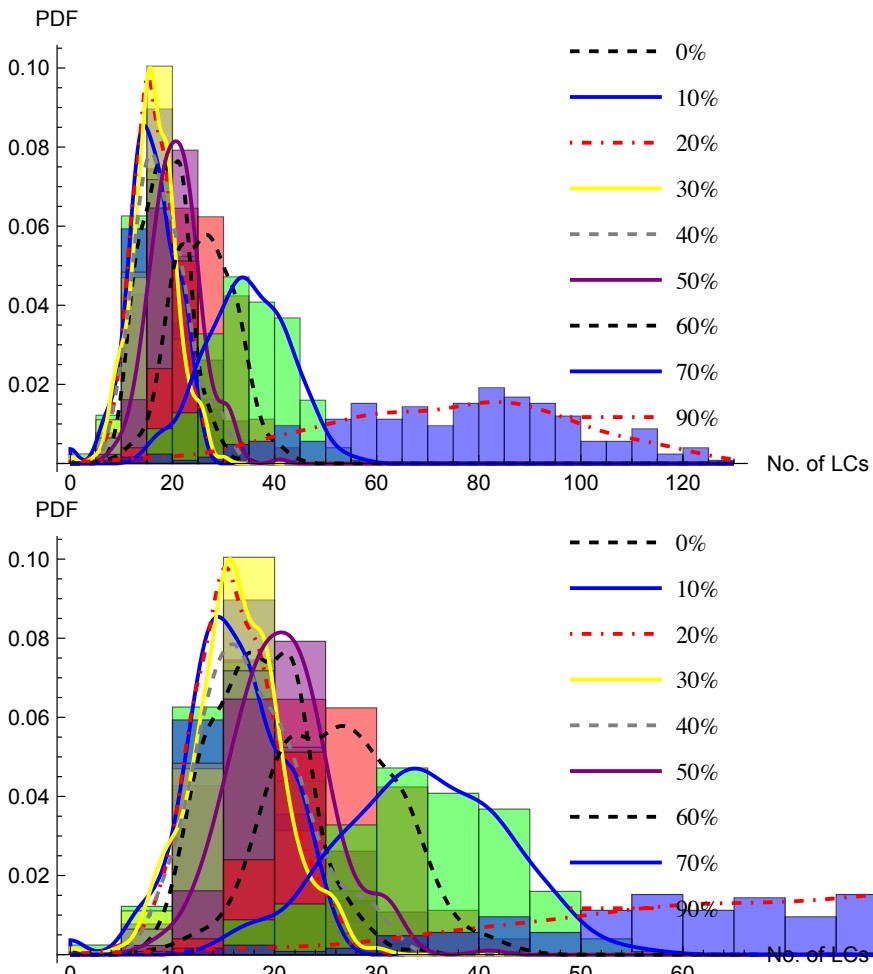

Figure 10: Extra data for the dropout tests.

| Dropout | Dropout | Result | p-value |
|---|---|---|---|
| 10% | 0% | Reject | 0.0000489944 |
| 10% | 10% | Do not reject | 1. |
| 10% | 20% | Do not reject | 0.690898 |
| 10% | 30% | Do not reject | 0.142326 |
| 10% | 40% | Reject | 0.0134828 |
| 10% | 50% | Reject | $3.9380742948255406 \times 10^{-17}$ |
| 10% | 60% | Reject | $1.2271320907480274 \times 10^{-46}$ |
| 10% | 70% | Reject | $1.9831398445539827 \times 10^{-100}$ |
| 10% | 90% | Reject | $1.6549287930539336 \times 10^{-133}$ |

Table 4: Kolmogorov-Smirnov hypothesis tests for the dropout experiments compared to a run with 10% dropout.

| Dropout | Dropout | Result | p-value |
|---|---|---|---|
| 20% | 0% | Reject | $4.749456190224837 \times 10^{-6}$ |
| 20% | 10% | Do not reject | 0.690898 |
| 20% | 20% | Do not reject | 1. |
| 20% | 30% | Do not reject | 0.786929 |
| 20% | 40% | Reject | 0.000721937 |
| 20% | 50% | Reject | $4.513284368522981 \times 10^{-19}$ |
| 20% | 60% | Reject | $2.0560842299464827 \times 10^{-58}$ |
| 20% | 70% | Reject | $2.2951948334897817 \times 10^{-110}$ |
| 20% | 90% | Reject | $1.2430043593276543 \times 10^{-146}$ |

Table 5: Kolmogorov-Smirnov hypothesis tests for the dropout experiments compared to a run with 20% dropout.

| Dropout | Dropout | Result | p-value |
|---|---|---|---|
| 30% | 0% | Reject | 0.0000247196 |
| 30% | 10% | Do not reject | 0.142326 |
| 30% | 20% | Do not reject | 0.786929 |
| 30% | 30% | Do not reject | 1. |
| 30% | 40% | Reject | 0.0012839 |
| 30% | 50% | Reject | $7.368476097588623 \times 10^{-17}$ |
| 30% | 60% | Reject | $3.04037573854777 \times 10^{-49}$ |
| 30% | 70% | Reject | $5.39288092428949 \times 10^{-91}$ |
| 30% | 90% | Reject | $3.1672113682285154 \times 10^{-123}$ |

Table 6: Kolmogorov-Smirnov hypothesis tests for the dropout experiments compared to a run with 30% dropout.

| Dropout | Dropout | Result | p-value |
|---|---|---|---|
| 40% | 0% | Do not reject | 0.137226 |
| 40% | 10% | Reject | 0.0134828 |
| 40% | 20% | Reject | 0.000721937 |
| 40% | 30% | Reject | 0.0012839 |
| 40% | 40% | Do not reject | 1. |
| 40% | 50% | Reject | $2.476782445216003 \times 10^{-9}$ |
| 40% | 60% | Reject | $9.405620742296745 \times 10^{-34}$ |
| 40% | 70% | Reject | $1.2328533479970469 \times 10^{-77}$ |
| 40% | 90% | Reject | $1.7803277971274416 \times 10^{-127}$ |

Table 7: Kolmogorov-Smirnov hypothesis tests for the dropout experiments compared to a run with 40% dropout.

## A.4   RESULTS FROM EXPERIMENT 9: ACTIVATION NOISE

We added in Gaussian distributed activation noise around a mean of 0.5 to all HLNs (which were sigmoidal, so the activation varied between 0 and 1), the results for three networks are shown in figure 11. The addition of noise changes the distribution LCs seen. For the overpowered NNs, those above the $n_{HLN}$ peak in experiment 1, the number of local codes increases significantly in most cases. For the underpowered NNs ($n_{HLN} = \{100, 500\}$) the number of local codes decreases, and this is not significant for tightly distributed noise.

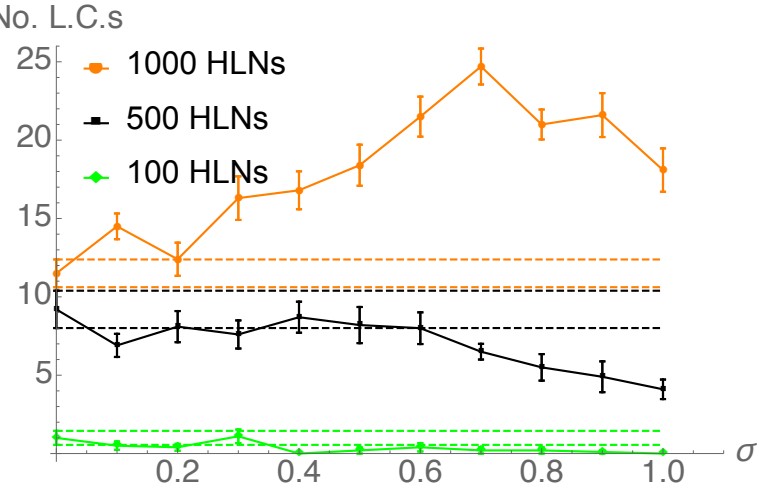

Figure 11: Including activation noise affects the number of local codes, $\sigma$ is the standard deviation of the Gaussian;y distributed noise. For NN with 1000 HLNs adding noise generally increases the number of local codes, for NN with 500 HLNs there is generally no effect or a decrease with high sigma. Lines are drawn to mark the standard error around the mean of LCs at $\sigma = 0$, i.e. no added noise.

## A.5   FURTHER DATA FOR EXPERIMENT 12: MULTI-LAYER PERCEPTIONS.

Examples of the modified MNIST input data are given in  8. In the invariant case a block of 20 (2.5%) hot pixels (fully on) are positioned around the edge of the image, thus a NN could learn to attend to the hot pixels in order to correctly assign the category instead of learning to identify the letter shapes. In the Gaussian case the prototype code perturbed by having the pixel value drawn from a different Gaussian distribution for each image (this is the equivalent of the perturbation of prototypes in experiment 7. The networks were trained with no regularisation and no validation set.

The results for the MLP experiment (11) is given in table 9. There is a small, but significant emergence of LCs in response to categories with added invariants, but it has to be for all images in a category for the NN to develop a local code to look for the invariant. In the Gaussian examples, no local codes emerge. Given that the invariants were less <2.5% of the input data, and the result that perturbation removes local codes above a certain level, we suspect that were the prototypes perturbed less or bigger, LCs might have been seen with the perturbed invariants.

Table 10 shows the impact on training times of the presence of a reliable code, which the NN can learn instead of learning to identify the number shapes. The NN trains much faster as the problem it is solving is a simpler one. Thus, the presence of an invariant adds a 'short-cut' to the task which the NN can find.

## A.6   FURTHER GENERALISATION RESULTS FOR EXPERIMENT 10: TRAINING TIME

Figure 12 shows the correlation between the number of local codes and classification accuracy and pattern matching accuracy on the test data. For NNs with HLNs to the left of the peak (i.e. $n_{HLN} <$ 1000) show a slight negative correlation, those at the peak or above show a slight positive correlation, although there is lots of spread in this data.

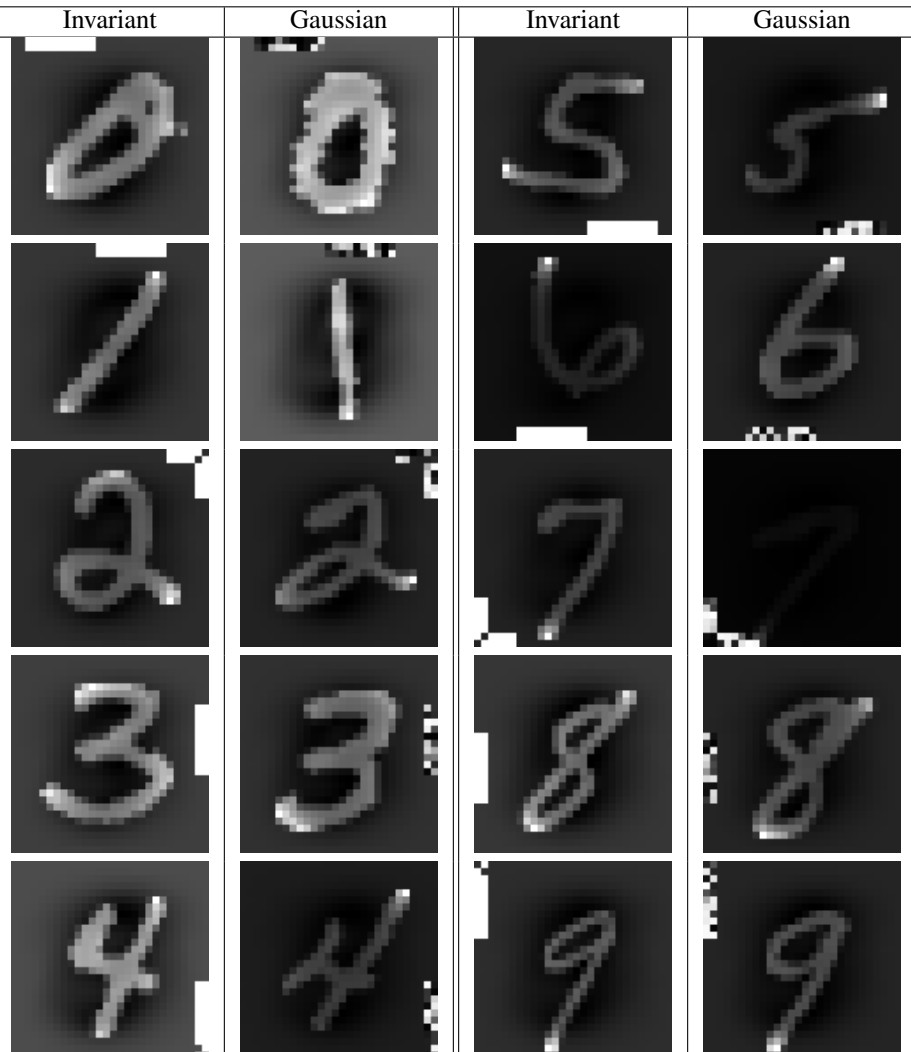

| Invariant | Gaussian | Invariant | Gaussian |
| --- | --- | --- | --- |

Table 8: Example hot-pixel invariant MNIST input data images (after normalisation from the mean image). 'Invariant' images have a hot-pixel short-cut code (of white pixels) that is the same for all examples of that class, 'Gaussian' invariant images have a greyscale hot-pixel code that is drawn from a Gaussian distribution and different for each image.

| data-set | Invariant | | | Gaussian | | |
| --- | --- | --- | --- | --- | --- | --- |
| | 48units | 100units | 500units | 48units | 100units | 500units |
| Standard | 0±0 | 0±0 | 0±0 | 0±0 | 0±0 | 0±0 |
| 50% all images | 0±0 | 0±0 | 0±0 | 0±0 | 0±0 | 0±0 |
| All images | 1.1±0.26 | 1.5±0.32 | 0.6±0.25 | 0±0 | 0±0 | 0±0 |
| 2 categories | 0.1±0.09 | 0.1±0.09 | 0.1±0.09 | 0±0 | 0±0 | 0±0 |
| 5 categories | 0.2±0.19 | 0.8±0.31 | 0.6±0.25 | 0±0 | 0±0 | 0±0 |
| 8 categories | 1.4±0.03 | 0.9±0.26 | 0.9±0.22 | 0±0 | 0±0 | 0±0 |

Table 9: Average numbers of LCs in 4-layer MLPs trained on MNIST (standard) and modified MNIST with added hot pixel invarients.

| data-set | Average no. epochs Invariants | Average no. epochs Gaussian distribution |
|---|---|---|
| Standard | 1090.8 | 1090.8 |
| 50% all images | 983.4 | 979.9 |
| all images | 221.7 | 460.9 |
| 2 categories | 1040.9 | 1025.0 |
| 5 categories | 494.3 | 537.0 |
| 8 categories | 327.5 | 462.3 |

Table 10: Training times for MLP networks with different MNIST training sets. Note that the presence of a invariant reduces training time because the NN finds the shortcut (all NN are trained to >99% acc), and introducing a GAussian distribution over that invariant (i.e. making it more variable) generally increases the training time.

| Accuracy | $n_{HLN}$ | gradient | intercept | $R^2$ |
|---|---|---|---|---|
| pattern matching | 250 | 25.7 | -0.18 | 0.23 |
| pattern matching | 500 | 68.8 | -0.01 | 0.0002 |
| pattern matching | 1000 | 130.7 | 0.82 | 0.26 |
| pattern matching | 2000 | 174.6 | 2.43 | 0.41 |
| classification | 250 | 26.5 | -0.08 | 0.23 |
| classification | 500 | 68.7 | -0.001 | $6.6 \times 10^{-6}$ |
| classification | 1000 | 123.0 | 0.42 | 0.34 |
| classification | 2000 | 144.0 | 1.33 | 0.50 |

Table 11: Fits for the trend lines drawn in figures 4 and 12.

## A.7 RESULTS FOR EXPERIMENT 13: SHUFFLING THE INPUT DATA

We tried shuffling the input data which should force the NN to memorise the dataset, rather than relying on the short-cut. We expected the number of local codes to drop to zero, but as shown in figure 13 this did not happen, instead the number of local codes was only reduced.

ACKNOWLEDGMENTS

An. Author would like acknowledge funding from Leverhulme on grant no.

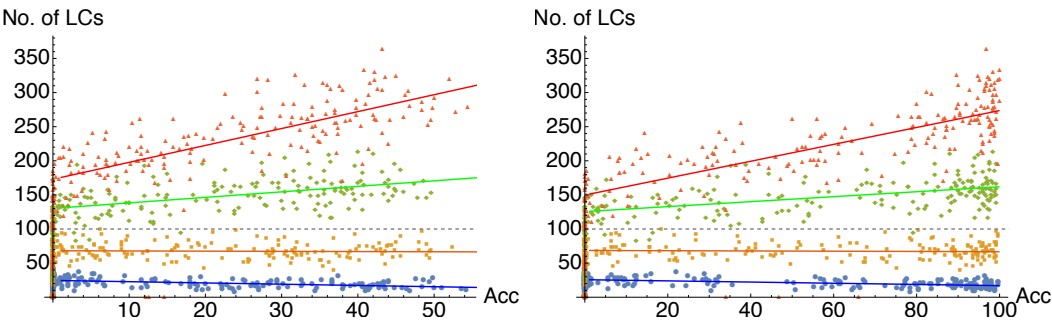

(a) The number of local codes increases with general-isation accuracy.

(b) The number of local codes increases with training accuracy.

Figure 12: Local codes are generally positively correlated with accuracy. Key: red: $n_{HLN} = 2000$; $n_{HLN} = 1000$; $n_{HLN} = 500$; $n_{HLN} = 250$.

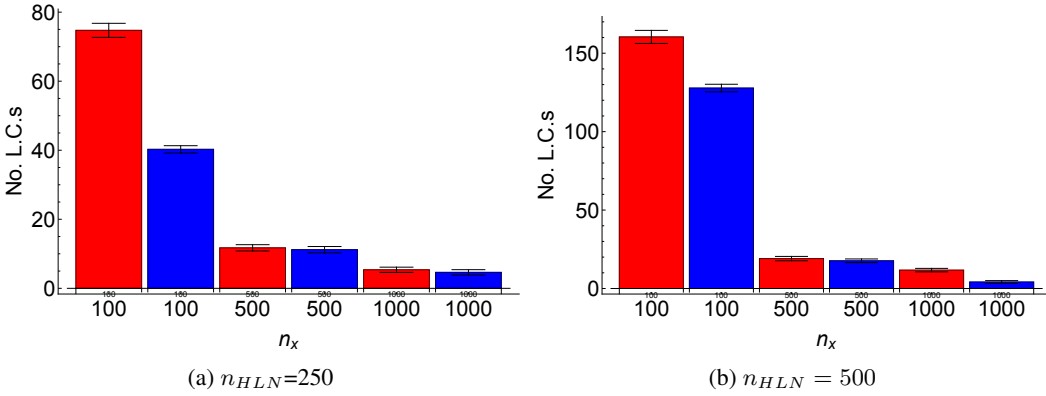

(a) $n_{HLN}$=250

(b) $n_{HLN} = 500$

Figure 13: Shuffling input data to create 'memorising networks' reduces the number of LCs but does not completely inhibit them. red: unshuffled data (control); blue shuffled input data.

