# OpenReview forum: "Under what circumstances do local codes emerge in feed-forward neural networks"
_ICLR.cc/2020/Conference — Reject_

### Official Review · AnonReviewer3 · 2019-10-21
**Official Blind Review #3**

**Rating:** 1

**Review:**

I have a lot of questions about the data used in the experiments. They are created according to the method explained in “Data design” (p.2). It is also summarized in the last paragraph of the first section as follows: ”there are 1/10 input bits that are always 1 for each class and these are the invariant bits, the 0s of each prototype are then filled in with a random mix of 1 and 0 of a known weight”. What is the intention behind this way of creating data? How general are the data created in this way as well as the analyses based on them? It seems to me that the data and thus the analyses lack the generality needed for the purpose of understanding behaviors of neural networks on real tasks/data.

The same is true for “Neural network design” (p.3), in which 13 experiments conducted in this study are explained. I think their explanations are too condensed; each explanation is very short and it is hard to understand the motivation and purpose of each experiment, i.e., what is the hypothesis to be verified and in what way it is verified?

In Experiment-12, MNIST is used as data unlike other experiments, and they are modified as “with added 20 pixel invariants”. What is the purpose of this modification? There is a statement in a footnote of p.5 “No LCs were seen in the standard MNIST runs”, which agrees with the above concern about the lack of generality.

Additionally, I do not understand the statement in p.5 “Increasing the difficulty of the problem (by increasing n_x, …”. Why does the use of more training data make the problem harder? It should usually be the opposite; the smaller, the harder.


**Experience Assessment:**

I have published in this field for several years.

**Review Assessment: Checking Correctness Of Derivations And Theory:**

N/A

**Review Assessment: Checking Correctness Of Experiments:**

I carefully checked the experiments.

**Review Assessment: Thoroughness In Paper Reading:**

I read the paper at least twice and used my best judgement in assessing the paper.

---

### Official Review · AnonReviewer1 · 2019-10-23
**Official Blind Review #1**

**Rating:** 3

**Review:**

The authors studied the local codes in neural networks through a set of controlled experiments (by controlling the invariance in the input data, dimensions of the input and the hidden layers, etc.), and identified some common conditions under which local codes are more likely to emerge.

The fact that local codes tend to emerge as a response to invariance is interesting but not surprising, especially given that convolution operations are designed to capture location invariance. It would be useful if the authors can clarify their contributions and compare against existing works in the literature.

Experiments are conducted at a relatively small scale: On a synthetic dataset with binarized vectors and on MNIST, which a predefined rule for noise injection (Figure 1). The controlled experiments conducted in the paper are still informative, but the overall message would be much stronger if the empirical analysis can be extended to common benchmarks such as CIFAR and/or ImageNet.

All of the experiments are based on very shallow networks (3-4 layers), and as the result, the study ignores batch normalization and skip connections which are common ingredients in state-of-the-art convolutional networks. It remains unclear whether the presence of those components would change the emergence behavior of local codes, and hence affect some of the conclusions in the paper.

**Experience Assessment:**

I do not know much about this area.

**Review Assessment: Checking Correctness Of Derivations And Theory:**

I carefully checked the derivations and theory.

**Review Assessment: Checking Correctness Of Experiments:**

I assessed the sensibility of the experiments.

**Review Assessment: Thoroughness In Paper Reading:**

I read the paper at least twice and used my best judgement in assessing the paper.

---

### Official Review · AnonReviewer4 · 2019-10-31
**Official Blind Review #4**

**Rating:** 3

**Review:**

Paper Overview:

This paper aims to study when hidden units provide local codes by analyzing the hidden units of trained fully connected classification networks under various architectures and regularizers.  The main text primarily studies networks trained on a dataset where binary inputs are structured to represent 10 classes with each input containing a subset of elements indicative of the class label.  The work also studies fully connected networks trained on the MNIST dataset (with the addition of some pixels indicating each class label).  After enumerating the number of local codes observed under these different settings, the authors conclude the following: (1) "common" properties of deep neural networks & modern datasets seem to decrease the number of local codes (2) specific architectural choices, regularization choices & dataset choices seem to increase the number local codes (i.e. increasing dropout, decreasing dataset size, using sigmoidal activations etc.).   The work then state that these insights may suggest how to train networks to have local codes emerge.

Review:
I particularly liked the simple dataset the authors construct in determining whether local codes emerge in hidden units, especially since deep networks and dataset used in practice are overly complex to gain insight for this behavior.  However, I find the overall message to be a bit confusing, especially in regard to using the analysis to construct networks with emergent local codes.  In particular, I feel that the authors could strengthen this work greatly by using their findings to train a deeper neural network for which local codes do emerge on a more realistic dataset.  Furthermore, this work would be significantly more impactful if a network with more local codes does generalize better, but that is unclear as of now (especially since local codes seem to not emerge in practical settings even though these networks are state of the art).

Criticisms/Questions:
(1) Main:  I'm somewhat confused about the main takeaway from this work in terms of understanding when local codes actually emerge in deep neural networks.  The authors seem to have a number of very specific conditions that are both architecture and dataset dependent, and overall I feel the message would be much stronger if the authors were able to rigorously study perhaps just a few of these conditions across many more settings.  For example, even just studying the impact of activation and providing some conditions/theory or a clearer understanding of which nonlinearities lead to more local codes would be insightful.  The current work seems to be more broad instead of tackling one of these properties in depth.

(2) I am a bit confused about the thresholds used by the authors in determining whether a hidden unit provides a local code or not. Do you just determine if there is some threshold given by the unit that separates out all points of one class from the rest?

(3) After several experiments, there are some heavy conjectures trying to rationalize the result of the experiment.  As an example, the authors provide statements like "ReLU is a more powerful activation function than sigmoid."  However, this statement in particular is not exactly correct, since given enough width, networks with either activation function should be able to interpolate the training data.  Another example of this is at the bottom of page 7, when the authors provide 5 possible explanations as to why local codes don't emerge in modern training settings.  It is unclear which of these explanations are true, but it would be great if the authors could actually provide a cleaner rationalization.

Minor criticisms:
(1) I've seen a number of different conventions for how to refer to the depths of networks, and I believe what you refer to as 3 layer networks would conventionally be referred to as 2-layer networks for theory audiences (as there are 2 weight matrices involved) or 1-hidden layer networks for empirical audiences.  I think adding a figure in the appendix for your architecture would clear up any confusion immediately.
(2) Some of the formatting is a bit awry: there are references to figures that appear as ?? (see page 8 paragraph 3).
(3) It would be nice to provide a consistent legend in some of the figures.  For example, Figure 4b has no indication for which settings the colors represent.
(4) As there seem to be a lot of experiments numbered 1-12, I think it would be much more readable to have different subsections on the different settings and outline the experiments in the subsection more clearly.  Referring back to these numbers on page 3 & 4 constantly makes it less readable.
(5) I quite liked Figure 8 in the Appendix.  I feel that this would have been a great figure to put towards the front of the paper to provide an example of local codes emerging.

**Experience Assessment:**

I have read many papers in this area.

**Review Assessment: Checking Correctness Of Derivations And Theory:**

N/A

**Review Assessment: Checking Correctness Of Experiments:**

I assessed the sensibility of the experiments.

**Review Assessment: Thoroughness In Paper Reading:**

I read the paper thoroughly.

---

### Official Review · AnonReviewer2 · 2019-11-04
**Official Blind Review #2**

**Rating:** 3

**Review:**

This paper studies the emergence of local codes in neural networks on a synthetic dataset. From my understanding, a neuron is counted as a local code if there is a class A such that its activations of data points from A are linear separable from its activations of data points from all other classes. However, is this definition for data points in the training set, or in the test dataset, or union of them? I did not find the exact definition in the paper.

It designed experiments to study the number of local codes. It made 7 empirical findings by the experiments on a synthetic dataset, listed in Section 1.1. It's findings are purely empirical. The authors may clarify this work's novelty and importance.

This paper seems to be finished in rush, because there is question masks, e.g., "Summary statistics and Kolmogorov-Smirnov hypothesis tests are reported in tables ?? in the appendix" in Page 8, "Results are shown in figure ?? and table 2." in Page 12, "As can be in seen tables ?? and figure ?? low values of dropout are likely the same distribution" in Page 12. The paper is very difficult to read for me, partly due to its writing in a language (local codes) that I'm not familiar with. I think that its presentation can be greatly improved for general audience.

I'm not familiar with the concept of "local codes", and I do not understand part of the paper.

**Experience Assessment:**

I do not know much about this area.

**Review Assessment: Checking Correctness Of Derivations And Theory:**

N/A

**Review Assessment: Checking Correctness Of Experiments:**

I assessed the sensibility of the experiments.

**Review Assessment: Thoroughness In Paper Reading:**

I made a quick assessment of this paper.

---

### Decision · Program_Chairs · 2019-12-19

**Decision:**

Reject

**Comment:**

This paper studies when hidden units provide local codes by analyzing the hidden units of trained fully connected classification networks under various architectures and regularizers. The reviewers and the AC believe that the paper in its current form is not ready for acceptance to ICLR-2020. Further work and experiments are needed in order to identify an explanation for the emergence of local codes. This would significantly strengthen the paper.